# Predictive Equations for Butterhead Lettuce (*Lactuca Sativa*, cv. Flandria) Root Surface Area Grown in Aquaponic Conditions

**Pamela A. Schwartz** [1],[†], **Tyler S. Anderson** [2] **and Michael B. Timmons** [2],[*]

1    Department of Statistics and Biometry, Cornell University, Ithaca, NY 14853, USA;
     pamela.schwartz75@gmail.com
2    Department of Biological & Environmental Engineering, Cornell University, Ithaca, NY 14853, USA;
     tyler.anderson@outlook.com
*    Correspondence: mbt3@cornell.edu
†    Current: Contract Software Developer, P&W Development LLC, Frankfort, KY 40601, USA.

**Abstract:** Aquaponic systems are becoming more prevalent and have led to accurate mass and energy balance models that allow nutrient utilization to be maximized and plant and fish systems to be coupled or complimentary. Such models still do not address the potential of using the plant side as both the primary nitrification system and as a sink for the nitrate being produced from the fish system. However, using the plants as the nitrification system for the fish waste requires a better understanding and quantification of the nitrification capacity of the plant system. A series of experiments were conducted using butterhead lettuce (*Lactuca sativa*, cv. Flandria) in deep water culture rafts. Plants were grown under two growing conditions and were evaluated based upon harvestable weight. Treatment 1 (H5) consisted of a standard hydroponic nutrient solution maintained at pH 5.8, while treatment 2 (A7) consisted of an aquaponic waste solution maintained at pH 7.0. The aquaponic conditions were created from a fish rearing system using koi (*Cyprinus carpio*) that was continuously recirculated between the designated plant tubs and the fish tank with an in-line bead filter to capture and mineralize fish solids. The total root surface area was not significantly different between treatments, but the ratio of root surface area to root fresh weight was different, suggesting that aquaponic roots are finer than hydroponic roots. Predictive equations were developed to correlate root surface area to shoot or root fresh weight, which can be used to design the nitrification component for a recirculating aquaculture system (RAS), as part of an integrated aquaponic system.

**Keywords:** replacement; inorganic media; biofiltration; coupled systems; root area to root mass ratio; nitrification

## 1. Introduction

Aquaponics is the combined culture of plants and fish where the unabsorbed fish feed nutrients are used to supply the nutrients needed by the culture plants. This is a marriage between a recirculating aquaculture system (RAS) for fish culture and a hydroponic system for plants. Maucieri et al. [1] has provided a recent review of hydroponic and aquaponic systems and has identified the key role that root area plays in the transport of nutrients into the plant. Goudriaan and Van Laar [2] wrote one of the early texts on modelling crop growth processes for traditionally grown agronomic crops (soil based). Others have written energy climate models that center around plants and energy balances [3]. Several researchers have published accurate mass and energy balance models that allow nutrient utilization to be maximized and plant and fish systems to be coupled or are complimentary [4–7].

While plant physiological processes will be fundamentally based upon the same principles, crop growth, and nutrient assimilation processes will be expected to be different under soilless hydroponic or aquaponic conditions. Surprisingly, researchers are finding that aquaponically grown lettuce even with high levels of sodium outperformed lettuce grown using conventional NFT hydroponic methods [8].

RAS employs biological filters for oxidation of ammonia that is excreted by the fish and is oxidized into nitrite and then nitrate, a relatively innocuous form of nitrogen. RAS biological filters are a significant capital cost in a RAS system. If traditional biofilters for the fish side of an aquaponic system could be replaced in functionality by the plant growing side, then the capital requirements for the fish system could be reduced. Additionally, supplemental income could be produced from the plant growing side and likely benefit from common marketing and infrastructure of the parent business. Currently, the design data to implement such a strategy is not available.

Equilibrium levels of nitrate-N in such systems can often exceed 100 mg/L depending upon the percentage of new water being added to the fish system. In some fish systems, denitrification may be required to lower nitrate levels to acceptable levels. While some species of fish are very tolerant of nitrate levels such as tilapia, other species such as the various salmonids often require nitrate-N levels less than 40 mg/L. One of the attractive features of aquaponics is that the plants utilize nitrate and when properly designed, there can be a balance achieved between production and utilization of nitrate. Timmons et al. [9] (see Chapter 19) provide a thorough review of aquaponic design.

RAS design includes a design step where the biological filter is sized using Equation (1) to accommodate the anticipated ammonia load $P_{TAN}$ (kg of total ammonia nitrogen per day) from the fish [9] (see Chapters 3, 7, and 8):

$$P_{TAN} = \frac{F \cdot PC \cdot 0.092}{t} \tag{1}$$

where '$t$' is usually one day, i.e., feed provided uniformly over a 24 h period, $F$ is feed provided per day, and $PC$ is protein content of the feed expressed in decimal form. Animal based proteins are generally assumed to be 16% nitrogen and the $PC$ is dictated by the species grown and management choices, but will generally be between 32% to 48%.

Once the total ammonia nitrogen (TAN) loading is calculated, sizing the biofilter is calculated either by an area (g of TAN per day per $m^2$) or a volumetric (g of TAN per day per $m^3$) basis. A 'rule of thumb' value given by Timmons et al. [9] for biofilter nitrification rates assumes that surface area rate (SAR) for nitrification is ~1 g TAN per $m^2$ per day. SAR rates are not known for plant roots, which can fully or partially serve as the biological filter of an aquaponic system. Media surface conditions and material properties will affect a SAR value as will TAN concentrations expected for the system water environment. Additionally, root age will impact SAR values, as bacteria require time to establish new populations as roots leave a system from crop harvest. As new baby plants are added and harvest sized plants are removed in a continuous harvest plant system, this mitigates a well know problem of reduced biofilter performance caused by collection of fish biosolids on the media.

Root SAR values are best determined by actual controlled tests to establish reliable design values. A first step in this regard is to quantify surface areas of root masses being generated for a particular plant, e.g., butterhead lettuce. Models have been presented that address the interactions between water and nutrient uptake and root growth [10]. In addition, mechanistic models of plant nutrient transport under hydroponic conditions that would be impacted by shoot/root mass dependencies are needed, since such models as reviewed by Rengel [11] and Malamy [12] only apply to soil based culture methods. Thus, the objective of our research was to determine root surface area as a function of lettuce shoot size as a first step in being able to predict nitrification capacity from an aquaponic system. A companion paper by Anderson et al. [13] determined nitrification capacity of a plant system coupled to a RAS.

## 2. Materials and Methods

Butterhead lettuce (*Lactuca sativa*, cv. Flandria) was grown under hydroponic or aquaponic conditions using five growing tubs made from HDPE polyethylene (1.82 m × 0.91 m × 0.30 m top dimensions of a tapered tub, 0.425 m$^3$) that were located in a glass greenhouse structure on the Cornell University, Ithaca, NY campus. Root surface area, fresh and dry weights of roots, and fresh and dry weights of lettuce shoots were collected from a single trial at harvest. Additional fresh weight data for shoots and roots were collected from three previous trials conducted using the same protocols and cumulative light over the course of their growing periods so that predictive growth curves could be developed [14]. Natural light was supplemented with artificial light to provide 15 mol/m$^2$/day. Lettuce seeds were planted in rockwool cubes in an ebb and flow bench and were watered four times equally spaced during daylight hours. At 11 days, 250 plants (50 per tub, equivalent to 30 plants per m$^2$) were selected for uniformity and then transplanted into each tub randomly. Styrofoam rafts 25 mm in thickness with 25 mm round holes were used to accept the plant plugs, which were spaced at 200 mm on center; rows were staggered to maximize uniformity of light to all sides of each plant. Lettuce was grown to harvest weights in these tubs for an additional 29 days to harvest (to reach a total of 595 mols/m$^2$ of light) for a total growth period of 40 days. Only plants from the interior locations were used for analysis to control for effects such as exposure to light, humidity, air movement, etc.

The two treatments differed in water source, nutrient source, and pH. Three tubs were subjected to normal hydroponic growing conditions (all inorganic salts), pH 5.8, and used reverse osmosis water, representing conventional hydroponic parameters, which will be referred to as treatment H5. Two tubs were subjected to aquaponic growing conditions, pH 7.0, and water sourced from a recirculating aquaculture system, referred to as treatment A7. The A7 treatment was created from a fish rearing system using koi (*Cyprinus carpio*) that was continuously recirculated between two of the plant tubs and the fish tank with an in-line bead filter in place to capture and mineralize fish solids. The A7 tubs received no supplemental nutrients other than chelated iron (Sprint 330), a necessary plant nutrient not found in fish feed, which was added initially to a concentration of 2 mg/L. The hydroponic nutrient solution (nutrient source for H5) was comprised of Sonneveld and Straver hydroponic lettuce formulation reduced to 50% concentrations of all nutrients and no silicon [15]. We used the 50% concentration as our research group had found no detectable performance difference with the lower concentration compared to the 100% formulation. Complete details of the experimental arrangement, nutrient concentrations, and growing protocol are provided by Anderson et al. [13].

Fresh shoot and root weights for all plants were collected at harvest. Plants were selected randomly from each tub for root fresh weight, root dry weight, and root surface area measurements. After harvest, roots were cut below the rock wool cube and stored in their respective nutrient solutions for two days at 5 °C and then roots were transferred to a 30% ethanol solution and continued to be stored at 5 °C [16] for twenty days to allow additional time to complete all surface area measurements. Storage had no apparent effect on the physical appearance of the roots. Roots were randomly selected each day for measurement from the pool of collected roots and root fresh weight recorded at that time.

The surface area for roots within the rockwool cube was calculated from measured length and diameter and assuming a cylindrical shape (diameter estimated as the average of the top and bottom diameters). The surface area of the roots outside of the rock wool cubes was determined using WinRHIZO Pro 2007 software (Regent Instruments Québec, QC, Canada) and an Epson Expression 10000 XL Scanner (Long Beach, CA, USA). The surface area of the root within the rockwool was determined to be less than 4% of the total root surface area and therefore dropped from the analysis for simplicity and because the root surface area within the rockwool cube is basically unavailable to support autotrophic nitrification. Dry-weights for roots and shoots were obtained after four to seven days in a drying oven held at 70 °C. Root surface area for the younger plants used to develop the growth curves was not collected, since their contribution to the total surface area would be minimal.

Since treatment groups had different sample sizes, non-parametric tests were used for analysis. The H5 treatment group was applied to three tubs of plants and the A7 treatment group was applied to

two tubs. Welch's *t*-test (unequal variances *t*-test) was used to identify response variable significant differences among treatment groups. Treatment effects on variables of interest to the producer (shoot weights and various ratios) and to the RAS system designer (root surface area, root surface area to shoot fresh weight, etc.) are summarized in Table 1.

**Table 1.** Overall mean (SD) and treatment means (SD) for each response variable with their coefficient of variations (CV = ratio of SD to mean). Root surface area is $cm^2$, all weights are in grams, and different superscript letters identify significant differences between treatments at α = 0.05. Average surface area and fresh weights of the lettuce root within the rockwool (not included in root measurements) were 28.1 $cm^2$ (SD = 9.02) and 2.80 g (SD = 0.720) respectively.

| Response Variable | H5 Sample Size | A7 Sample Size | H5 Mean (SD) | A7 Mean (SD) |
|---|---|---|---|---|
| **Shoot FW** | 71 | 48 | **187** [A] (22.4) | **166** [B] (18.0) |
| CV Shoot FW | | | 0.120 | 0.108 |
| **Shoot DW** | 35 | 22 | **7.61** [A] (0.712) | **6.92** [B] (0.619) |
| Shoot DW/FW | 35 | 22 | 0.0395 [A] (0.00303) | 0.0420 [A] (0.00613) |
| Root FW | 64 | 44 | 7.85 [A] (1.67) | 7.22 [A] (1.74) |
| Root DW | 30 | 16 | 0.354 [A] (0.877) | 0.333 [A] (0.111) |
| CV Root FW | | | 0.213 | 0.241 |
| Root FW/Shoot FW | 30 | 14 | 0.0425 [A] (0.00588) | 0.0452 [A] (0.01550) |
| Root DW/Shoot DW | 11 | 4 | 0.0418 [A] (0.0176) | 0.0394 [A] (0.0033) |
| Root SA | 12 | 8 | 728 [A] (107) | 743 [A] (224) |
| CV Root SA | | | 0.147 | 0.301 |
| **Root SA/Root FW** | 12 | 8 | **99.6** [A] (10.3) | **124** [B] (14.3) |
| Root SA/Shoot FW | 12 | 8 | 3.76 [A] (0.492) | 4.25 [A] (1.05) |
| CV Root SA/Shoot FW | | | 0.131 | 0.247 |
| Root SA/Shoot DW *p* = 0.21 | 12 | 5 | 94.7 [A] (11.8) | 127 [B] (47.8) |

## 3. Results and Discussion

Within a treatment, mean responses for shoot or root for the entire data set and the subset sample for shoot FW (*p*-value of 0.22) and root FW (*p*-value of 0.07) were not significantly different. For this reason, data from the large data set were used to develop predictive equations relating fresh shoot weight and root surface area. However, we only used the root area data from the aquaponic treatment, since the focus for our model is to predict root area for aquaponic growing systems. Results are presented in Table 1 on the physical characteristics of the lettuce plants for the two treatments (H5 is normal hydroponic and A7 is aquaponic).

It is apparent from the data in Table 1 that treatment had a significant effect on lettuce shoot fresh weight (H5 187 g versus A7 166 g). The elemental nutrient concentrations provided by the A7 treatment are generally reduced compared to the modified Sonneveld conditions (H5) and one would expect inferior growth performance [13]. Further compromising the A7 treatment is that pH 7.0 is not optimal for nutrient availability. Similar experiments were conducted by Anderson et al. [13] and they found aquaponic plant biomass responses were not different from hydroponic in all biomass response categories, even though the aquaponics system was operated intentionally at pH 7.0 (compared to the hydroponic system at pH 5.8) and had non-optimal nutrient solution elemental concentrations.

A similar experiment conducted by Nozzi et al. [17] found that lettuce shoots derived from aquaponics (nutrients provided only from fish with no supplemental iron) were 29% smaller in size than their hydroponic treatment. In our experiment where supplemental iron was provided, we saw a reduced effect of 11% smaller shoot size for aquaponic shoots (Treatment A7, nutrients only from fish and supplemental iron) when compared to hydroponic shoots, suggesting that supplementing for iron dramatically increased shoot fresh weight in an aquaponics system. Nozzi's experiment using deep water culture tanks was single pass systems and the pH's were 6.0 and 6.5 in the hydroponic and aquaponic treatments, respectively. While the higher pH in the aquaponic treatment could account for some compromise in nutrient availability, particularly iron, the addition of iron to the aquaponic

solution had dramatic positive effect on the aquaponic lettuce yield. This supports Liebig's Law of the Minimum which states that yield is proportional to the amount of the most limiting nutrient, whichever nutrient it may be. (A/N: Justus von Liebig (1803–1873) was a German chemist and pioneer in organic chemistry; he first published on agricultural chemistry in 1840).

Root FW was not significantly different between treatments at $\alpha = 0.05$. However, Root FW was significantly different at $\alpha = 0.10$. Treatment had no significant effect on mean responses among treatments for root surface area at $\alpha = 0.05$. However, the coefficient of variation (CV) in root surface area increased from 0.147 in the H5 treatment at pH 5.8 to 0.301 in the A7 treatment at pH 7.0, suggesting greater variability in the aquaponic treatment. Interestingly, root surface area to root fresh weight ratios from the hydroponic treatment (H5) was significantly smaller than the aquaponic treatment (A7).

The larger aquaponic CV value for the ratio of root surface area to root fresh weight compared to the hydroponic value could be an indicator of environmental or nutrient stress suggesting the A7 aquaponic treatment responded by increasing root mass and area in response to the non-ideal conditions. Supporting this conjecture is that we found the root surface area to root FW ratio for A7 (124 cm$^2$/g) was significantly larger than the H5 (99.6 cm$^2$/g) at $\alpha = 0.05$. This suggests that the A7 group had finer roots relative to the H5 treatment, which is mathematically supported by the data collected. The ratio of root surface area to shoot fresh weight was not different. Once again, for this response variable, the coefficient of variation increases for the aquaponic treatment A7 compared to the hydroponic treatment H5. The A7 root surface to root mass ratio was higher than the H5 treatment. This means that for a given mass of roots, there is more surface area that would happen if the aquaponic roots are 'finer' in diameter. This hypothesis is supported by the photos of roots from both treatments, see Figures 1 and 2.

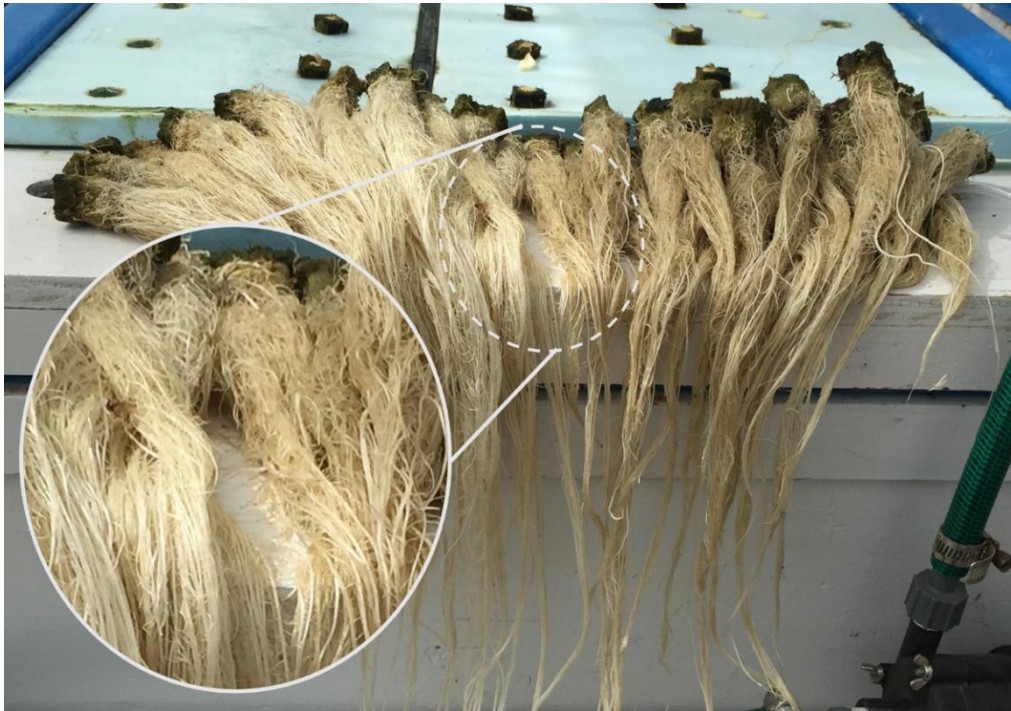

**Figure 1.** Individual root masses collected from hydroponic (on left) or aquaponic (on right) deep water culture systems. (raft can be seen in background with the rock wool plugs still in place after cutting off the shoot).

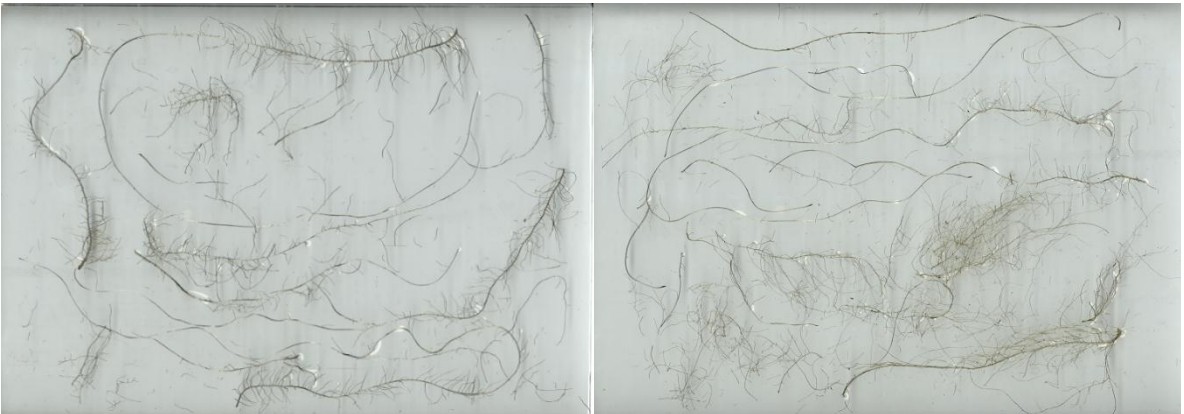

**Figure 2.** A magnification of individual lateral root strands from either hydroponic (on left) or aquaponic (on right) grown lettuce using deep-water culture methods (hydroponic treatment has the bold upper left root lateral shown).

It is clear that there is some difference in root development and characteristics between ideal hydroponic conditions and aquaponics conditions as seen in the data in Table 1. Others have drawn the same conclusion [8]. Yogev et al. [4] found in NFT systems that shoot weight in nutrient enriched aquaponic water showed a significant growth rate increase compared to the hydroponic or non-supplemented aquaponic water treatments, yet the root weight was similar between both aquaponic water treatments, and both were significantly higher than that observed in the conventional hydroponic treatment and further postulated that microorganisms and dissolved organic matter played an important role in RAS water for promoting plant roots and shoots growth. Recent research by Goddek et al. [6] found that lettuce plants grown in a hydroponic NFT system supplied with supernatant from an anaerobic reactor had significantly better performance with respect to weight gain than those in an NFT system where supernatant from an aerobic reactor was added. Goddec et al. hypothesized that the effect was caused by the presence of $NH_{4+}$, as well as dissolved organic matter, plant growth promoting rhizobacteria and fungi, and humic acid, which are predominantly present in anaerobic effluents.

From an engineering design perspective, the aquaponic conditions would provide some safety factor to the design engineer, since a bit more surface area (2% more) is created for a given amount of lettuce shoot biomass. Thus, applying the equations that are developed below can be applied confidently.

*Predictive Regression Equations*

The primary objective of this research was to provide predictive equations of root surface area as a function of lettuce shoot weight. There is limited data on root area related to root mass, since these parameters are basically of no interest to growers who are interested in how much saleable lettuce (shoots) can be produced. Lettuce shoot weight is commonly designed for and is typically reported as the harvested product for market. The surface area is needed by the designer of the aquaponic system to determine nitrification rates to support a design fish load.

Predictive regression equations are developed below followed by an example to demonstrate how the equations can be used to determine the amount of TAN the biofilter from the hydroponic system can accommodate in a day. The data used to develop the predictive equation for root surface as a function of root fresh weight is from the A7 treatment only since we are interested in the effect of aquaponic growth conditions on root surface area. The data used to develop the predictive shoot fresh weight equation as a function of root fresh weight is from the H5 treatment only; we consider this acceptable since our comparison of H5 vs. A7 showed no significant difference between these two treatments for root fresh weight to shoot fresh weight ratio.

A linear regression was used to predict root surface area as a function of root fresh-weight for aquaponic conditions A7 (Figure 3 and Equation (2)). We used a linear model, since we had no basis argue that a higher order model better represented the biological process plant growth. Note that a polynomial term–a quadratic (squared) or cubic (cubed) term (our 'x' is $FW_{Root}$) turns a linear regression model into a curve. But because it is X that is squared or cubed, not the Beta coefficient, it still qualifies as a linear model. This makes a straightforward way to model curves without having to model complicated non-linear models.

The regression equation was found to be significant (F (1, 6) = 53.7, $P$ = 0.00033), with an $R^2$ of 0.90 and an adjusted $R^2$ of 0.88 (Equation (2)). Surface area increased 94 cm$^2$ for each gram of root fresh-weight. Some modelers may develop lettuce growth models starting from seed when there is no shoot mass, so we also provide a model where the intercept is forced to zero. For this case, the slope is ~119 cm$^2$ per gram of root fresh-weight; the equation is not shown.

$$SA_{Root} = 169 + 94 * FW_{Root} \tag{2}$$

where

$SA_{Root}$ = Surface area of roots, cm$^2$
$FW_{Root}$ = root fresh-weight, g

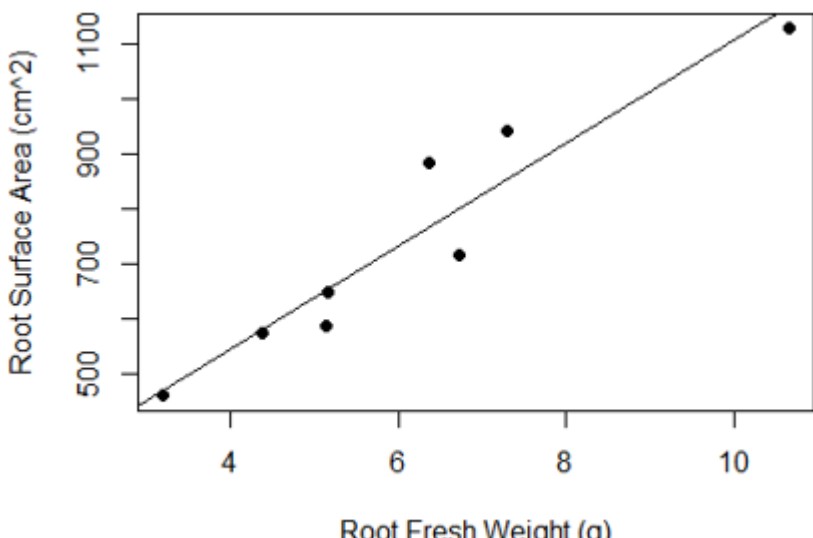

**Figure 3.** Root surface area vs. root fresh weight for aquaponic treatment only (root fresh weight excludes weight in the rockwool cube).

The data for fresh shoot and root weight collected from H5 and A7 at various times throughout the growth period was used to develop a predictive equation using a Gompertz growth model [18]. We used fresh weight of the root ($FW_{Root}$) to represent the time variable in the Gompertz equation (Equation (3)), since the plants were collected at increasing days of age while employing a fixed light level per day; the majority of the data was collected at harvest. The reason we did not use time (age) is that growth is dependent upon light received per day and not age or time. Thus, time is represented by the size of the shoot weight and associated root fresh weight. Our data is consistent with earlier work published from the same lab [19,20].

$$y(t) = \alpha e^{-\beta k^t} \tag{3}$$

where

$\alpha$ = upper asymptote

$\beta$ = growth displacement

$k$ = growth rate

$t$ = time variable

The resulting model gave significant ($p < 0.05$) parameters; = 186 (SD = 2.70), $\beta$ = 34.1 (SD = 14.4), and $k$ = 0.409 (SD = 0.0381) (Equation (4) and Figure 4.)

$$FW_{Shoot} = 186e^{-34.1(0.409)^{FW_{Root}}} \tag{4}$$

where

$FW_{Shoot}$ = shoot fresh-weight, g

$FW_{Root}$ = root fresh-weight, g

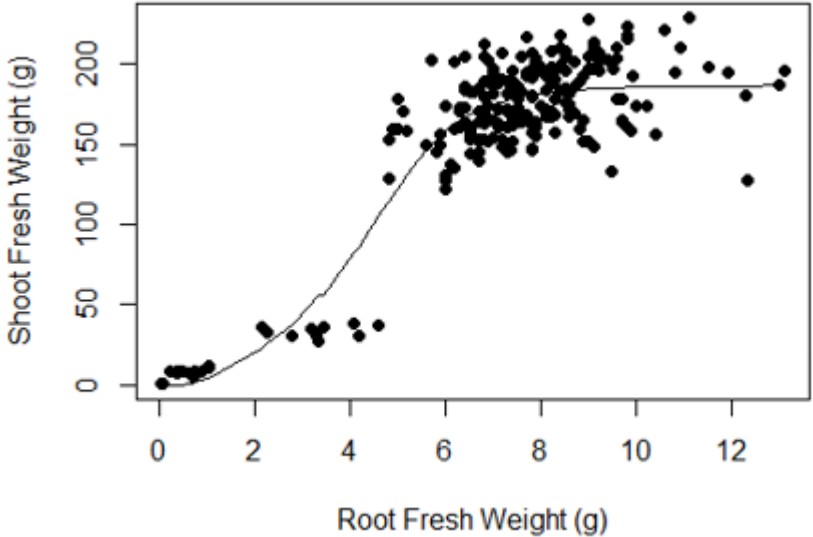

**Figure 4.** Gompertz growth curve of shoot fresh-weight vs. root fresh weight.

A simple linear regression was also done to predict shoot fresh weight on root fresh weight (Figure 5). The regression equation was significant (F (1, 220) = 450.5, $P < 2.2 \times 10^{-16}$), with an $R^2$ of 0.67 and an adjusted $R^2$ of 0.67 (See Equation (5)). Shoot fresh-weight increased 18.1 g per gram of root fresh-weight. If the intercept is forced to zero, then the slope is ~21.8 g per gram of root fresh-weight (equation not shown). The Gompertz model (Equation (4)) compared to the linear model (Equation (5)) predicts heavier shoot fresh weights as roots increase in weight above 5 g, which in turn would then predict larger root surface areas. The linear model would predict smaller or more conservative surface areas as a function of shoot fresh weight and might be preferred for this reason for design purposes.

$$FW_{Shoot} = 29.3 + 18.1 * FW_{Root} \tag{5}$$

where

$FW_{Shoot}$ = Fresh-weight of shoots, g

$FW_{Root}$ = root fresh-weight, g

Engineers and plant scientists should find the above equation useful since it now provides a simple equation to predict lettuce shoot weight as a function of root mass. This might seem odd since the basic objective of the paper is to predict root area as a function of root mass. The reason we constructed the model this way is because the majority of data in the literature always provides lettuce

shoot weight (since that is the marketable product), but data for root mass is much more rare. Using Equation (5), one can iteratively solve what root mass is needed to match up with the known shoot fresh weight. Additionally, having a root mass value, one can use Equation (2) to predict root surface area. As additional data is collected by researchers in this area, such data could be contributed to an open-source greenhouse modelling platform, as created by Körner [21] as these types of models are further refined.

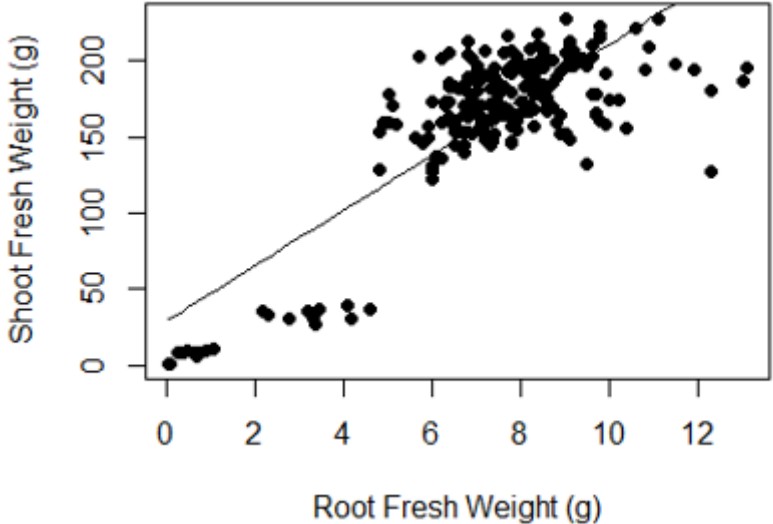

**Figure 5.** Regression of shoot fresh weight on root fresh weight.

## 4. Conclusions

Based upon lettuce root weights, shoot weights, and root surface areas collected from hydroponically grown lettuce using deep pond floating rafts grown under hydroponic or aquaponic conditions to commercial harvest weights, the following conclusions were drawn:

1. Treatment had a significant effect on lettuce shoot fresh weight.
2. Root surface areas were not different between treatments.
3. Root surface area to root fresh weight ratios from the hydroponic treatment (H5) was significantly smaller than the aquaponic treatment (A7), suggesting that the aquaponic roots were finer.
4. Predictive equations can be created to predict root surface area as a function root fresh weight.

**Author Contributions:** M.B.T. conceived the project. M.B.T., T.S.A., and P.A.S. designed the experiment; P.A.S. collected and analyzed the root surface data; T.S.A. conducted the experiment. All three authors wrote the paper.

**Funding:** This research was supported entirely by the Cornell University Agricultural Experiment Station federal formula funds, Project No. 1237650 and NYC-123421 received from Cooperative State Research, Education, and Extension Service, U.S. Department of Agriculture.

**Acknowledgments:** We would like to thank Erica Cartusciello and Jasmine Williams for their early assistance in developing our area measuring techniques. Finally, the paper was significantly improved by the thoughtful suggestions of the MDPI reviewers and Jonathan Allred, a candidate at Cornell University.

**Conflicts of Interest:** The authors declare no conflict of interest. Any opinions, findings, conclusions, or recommendations expressed in this publication are those of the author(s) and do not necessarily reflect the view of the U.S. Department of Agriculture.

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
