# Peer review of "Predictive Equations for Butterhead Lettuce (Lactuca Sativa, cv. Flandria) Root Surface Area Grown in Aquaponic Conditions"

_horticulturae, doi:10.3390/horticulturae5020039_

Round 1

Reviewer 1 Report

good work and nicely written.

Author Response

Response to Reviewer #1. We incorporated and addressed all suggestions and comments. In the track change version of the manuscript, I added comment addressed to Reviewer 1 to show her/him where the corrections were made. He/She made really good suggestions (not many but ON TARGET). Thank you.

Reviewer 2 Report

Line 13-14: cannot support this statement. also not part of the intro. Needs to be added. There are many models out there that provide good tools for sizing such systems.

The introduction does not comprise recent research with respect to:
(1) sizing aquaponics systems
https://doi.org/10.3390/w8120589
https://doi.org/10.1016/j.aquaeng.2018.07.001
https://doi.org/10.1016/j.agsy.2019.01.010

(2) root-shoot observations in AP systems:
https://www.mdpi.com/2073-4441/8/10/467
https://doi.org/10.3390/agronomy6020037

Should be added to introduce the topic.

line 71: punctuation mistake.

line 71 & 72: this statement is inaccurate. The root development and nutrient uptake rate in soil (as stated in source #4) is totally different than under hydroponics conditions. Also, nutrient uptake as well as root development depend on several factors such as:
(a) relative humidity
(b) (global) radiation
(c) temperature
(d) nutrient availability and ratio
(e) bacterial composition in, or sterility of the hydroponics nutrient solution

Relevant sources are:
https://doi.org/10.17660/ActaHortic.2017.1154.32
https://doi.org/10.21273/hortsci.42.2.272
https://doi.org/10.1007/s10499-018-0293-8
https://www.springer.com/de/book/9780792332190
https://doi.org/10.3390/w8120589

line 152: exactly. That pH inhibits nutrient uptake. Or at least the uptake of specific nutrients. Also I am wondering about the nutrient composition of the A7 treatment. Were all ratios as they are supposed to be for lettuce? (see: https://www.taylorfrancis.com/books/9781439878699)

line 154: Valentina Nozzi was indeed conducting this experiment. However, she conducted it in a one-loop systems. So what was the pH in her experiment; please discuss this. What about Liebig's law of the minimum in case one element is scarce? Please ALSO refer to her actual publication and not her conference presentation (https://doi.org/10.3390/agronomy8030027).

line 290f: Nice example, but there is so much more to discuss. First of all Racozy does not publish in peer-reviewed journal. So he basically can claim whatever. In addition, you are comparing apples and oranges. The environmental conditions are crucial factors for nutrient uptake. Even microbial communities in the water, etc. Also the pH in each system is different. Just compare the pH in decoupled multi-loop systems to the one in a conventional one-loop system. You will find totally different root and shoot growth observations. You need to discuss this much better and also refer to more external literature. 12 references are by far not sufficient for such a publication. Dive into the literature and discuss your findings creatively.

line 306: What does that actually tell us? :)

line 309: Why? What do you think is the reason?

line 312: Be more critical on that statement. In my opinion, environmental parameters should be part of the equations to make it actually useful.

Author Response

Reviewer 2. Reviewer's comments and questions were extremely helpful. He/She also provided links to references that supported/refuted points in the paper. ALL were incorporated into the paper. Thank you.  For all the other reviewer comments, each was addressed in the paper.  I enter/revise text and then I enter a comment that starts out Reviewer 2.

One of the reviewer's last comments is particularly important and I want to highlight it here:

Reviewer 2 made the following comment: ine 312: Be more critical on that statement. In my opinion, environmental parameters should be part of the equations to make it actually useful.  RESPONSE.  We totally agree that environmental conditions will affect shoot, root mass and root surface area. However, the objective of this paper is to have the predictive equations for ‘normal’ aquaponic water quality conditions. These were the conditions under which our data was collected, e.g, our lettuce from seed to harvest in ~35 days and heads that were ~ 150g (thus normal).  So, I think our Conclusion #5 is valid.

Again, this reviewer was very very helpful. Constructive criticism is always good.  Thank you. MB TImmons.

Reviewer 3 Report

The present manuscript fit within the general scope of the Journal. The authors aimed to compare by using specific model the root surface area of an important leafy vegetables such as lettuce grown in two systems hydroponics and aquaponics. The topic is not novel since a lot of published data are present in the scientific literature. The paper seems more a technical rather than a scientific paper.

I have several major concerns:

The abstract section should be re-written in particular the results highlights where the differences between treatments should be reported in percentage in order to be clearer for the readers.

The authors should avoid to use the same keywords already present in the title.

The hypothesis in the Introduction section just before the objectives should be inserted. In other words, what is the novel aspect of the current paper in comparison to the previous published papers.

The Results and discussion section should be separated in order to avoid redundancy.

The discussion section is the weak point of the paper and it is completely missing probably due to the limited data set presented. Why only the root and shoot biomass were measured, what about the mineral status and the physiological status?

The conclusion section in a scientific paper should not report as numbers it seems more like a Technical reports.

The paper could not be accepted in the current form but it could be resubmitted as a new manuscript after major revisions and by adding more data to support the conclusions.

Author Response

Reviewer #3.

I made significant revisions based upon the comments of Reviewer #3. These changes are highlighted in the text by me highlighting a section and then inserting a comment, addressed to Reviewer #3.

But, I did Reviewer #3 after having addressed the comments and suggesitons of Reviewer #1 and #2. These changes made significant text changes.

As a result, I entered some comments (beginnng) to make some over-all explanations of how the text was changed to address commonly identified concerns.

I even changed the TITLE to highlight what the focus of the paper is.

Thank you for your suggestions.  Much better paper as a result of your help. THank you.

Round 2

Reviewer 2 Report

Thanks for incorporating my suggestions. I wish you all the best for your further research. :)

Reviewer 3 Report

The manuscript has improved with new revisions, so I accept it in the present form.